# Spatiotemporal Change of Urban Sprawl Patterns in Bamako District in Mali Based on Time Series Analysis

**Moussa Aliou Keita [1],\*, Renzong Ruan [2] and Ru An [2]**

1   School of the Earth Sciences and Engineering, Hohai University, Nanjing 211100, China
2   School of the Hydrology and Water Resources, Hohai University, Nanjing 211100, China;
    rrz@hhu.edu.cn (R.R.); anrunj@hhu.edu.cn (R.A.)
\*   Correspondence: Moussa@hhu.edu.cn

**Abstract:** For decades, urban sprawl has remained a major challenge for big cities in developing countries, such as Bamako. The aim of this study is to analyze urban sprawl pattern changes over time in the Bamako district using landscape index analyses. Four thematic maps of land cover (LC) were produced by applying the maximum likelihood supervised classification method on Landsat images for 1990, 2000, 2010, and 2018. Five landscape indexes were selected and calculated at class level and landscape level using FRAGSTATS software. The results showed that the dominant class for all the years within the landscape was a built-up class. Forest class covered the smallest area in terms of the percentage of land (%PLAND), and was the weakest class in terms of number of patches (NP) and largest patch index (LPI). Grassland is defined as the class with the highest fragmentation, farmland with the highest shape irregularity and more heterogeneity, and built-up with the highest patches. Class area (CA) of built-up showed the importance of sprawl in Communes 6, 5, and 4, respectively. Indices trends and land use/cover showed infill, scattered, and ribbon developments of sprawl. This study contributes toward monitoring long-term urban sprawl patterns using index analyses.

**Keywords:** urban sprawl patterns; Bamako district; landscape index; patterns change; Mali

## 1. Introduction

Urbanization and urban development are keys toward global economic growth in the 21st century [1–4]. These processes show no sign of slowing down, and are probably the most powerful, unforced, man-made forces that have emerged from fundamental change in urban land cover and landscape around the world [5].

In recent years, urban sprawl pattern analysis has become an important field of research around the world, mainly in developing Asian countries where different aspects about urban sprawl were studied. While some studies have explored the economic, demographic, and natural implications of urban sprawl processes [6–8], others focused on political, environmental, and others aspects, such as the impacts [7,9,10]. However, in recent years, urban sprawl has been studied in China and India, more so than other places. Most of these studies, using different methods and approaches, found almost the same results concerning the causes, characteristics, and processes of urban sprawl. The demographic, economic, natural, and political aspects are considered the mains factors of urban sprawl [1,11,12]; while ribbon development, scattered development, and the infill development are the main forms of urban sprawl patterns [13–15]. Arable loss, urban pollutions, and natural hazards (e.g., flooding) are the main impacts of urban sprawl [16–18].

There are many "landscape" terms for the landscape. The definition of "one-in-one" allows it to communicate clearly, and to formulate a "Manage" policy. It is a plot area that contains a patch, mosaic, or landscape element. Landscape is a heterogeneous land that forms a cluster of icing ecosystems in the form of re-formation [19]. The concept is different from the normal ecosystem vision, with a focus on ecosystem groups and their interactions.

There are other definitions of landscape, in terms of research or management background. For example, from a wildlife perspective, we can define it as a mosaic land containing habitat patches, where "ten-in-one" or "target" habitat patches are embedded [20]. Landscape is a heterogeneous model of an area of the physical world where certain properties of the environment have scarlet as linear features, plaques, points, or continuous variation surfaces in space. For example, planning landscapes are typically watersheds and management areas measured in tens to thousands of acres. In contrast, ecological research focuses on moldology or a few square feet (a few square feet) of common interest. The size and scale of the landscape is a direct function of its purpose. In the area of habitat monitoring, the landscape must also reflect a meaningful spatial range and food for the emphasis species [21].

In geographical terms, landscapes are defined as the combination of environmental and human phenomena, which coexist in specific locations on the earth's surface. Urban areas are the most striking example of the human landscape. These areas involve the highest levels of human activity and are often severely affected by environmental factors. Remote sensing data and technology, combined with GIS and landscape pointers, are helpful to study such landscapes.

This dais is the basis for analyzing and describing land cover (LC) and its changes [22]. Remote sensing images contain a wealth of information regarding morphology, composition, and dynamics of urban areas. It was widely proven to be a reliable means of urbanization [23,24]. Collecting information about changes in LC is essential to better understand the relationships and interactions between humans and the natural environment. Remote sensing data is one of the important data sources for LC space–time and LC change research [22], and the use of remote sensing technology shows that urbanization has become an important requirement of research. Since it explains the correct use of remote sensing pointers, it is extensively used by researchers in urbanization analysis [24].

Many studies were carried out using remote sensing data sets to study changes in space–time landscape patterns [16,25–27]. The main purpose of these studies is to analyze the LC dynamics of space–time, especially urban growth/disorder and rural land loss. Most of these studies clearly show that LC patterns and their changes are related to natural and social processes [22]. These natural and social processes, known as change factors or drivers, may be related to changes in physical conditions in the landscape environment, natural disaster events, economic growth, population growth, political management, etc. Therefore, the development of dedicated GIS and remote sensing (RS) technologies is very clear in analyzing LC changes and understanding the dynamic stakes that can drive land conversion in urban and rural areas [22]. Landscape pattern index is widely used to study the spatial characteristics, change analysis, urban land use driving forces, and simulations to predict future urban spatial patterns [6,15,18,28].

Many studies have also focused on landscape pattern analysis methods. A standardized approach to measure and monitor landscape pattern attributes is described to support habitat monitoring [1]. The process of monitoring uses disaggregated landscape maps, where selected habitat attributes or different categories of habitat quality are represented as different patch types, using maps generated by modeling methods [29]. The term "landscape pointer" is often used only to refer to indexes developed for classified maps. In addition, although most landscape pattern analyses involve the identification of pattern proportions and intensity, landscape pointers focus on the representation of the geometry and spatial properties of classified map patterns on a single scale [30].

Many studies have also focused on urban sprawl spatiotemporal landscape using remotely sensed data [10–14] in different aspects, such as characterizing sprawl patterns [7–9,11], predicting sprawl pattern changes in the future using regression models [13,17,31,32], and quantifying sprawl patterns [16,33,34]. Most of these studies have used different methods to achieve their purposes (such as spectral, indexes [35], regression models, cellular automata Markov chain (CA-Markov) model, multi-approach analysis, etc.), in order to characterize, predict, and quantify the urban sprawl pattern [23,31,36]. According to these studies, urban sprawl patterns are

characterized by three main types: ribbon pattern, scattered pattern, and leapfrog pattern. These patterns are supported by three processes of sprawl: linear development, scattered development, and infill development [24,25].

The Bamako district is retained for this study because, on the one hand, it is the main and the biggest city in Mali; therefore, it faces a faster urban sprawl process, causing multiple socioeconomic and environmental issues. On the other hand, no study has investigated urban sprawl in the Bamako district, specifically by using landscape metrics that understand urban sprawl. Thus, it will give new insight into Mali, in general, and the Bamako district, in particular.

The main purpose of this study is to analyze landscape pattern changes in the Bamako district by using four satellite images from four years (1900, 2000, 2010, and 2018), which were acquired and processed with the supervised classification to create LC maps. Based on the produced LC maps, the calculation of spatial metric indexes, using FRAGSTATS software, is also a key part of this work. The selected spatial metrics are: the percentage of land (%PLAND), the number of patches (NP), the landscape shape index (LSI), the largest patch index (LPI), and the contagion index (Contag). The landscape indexes are used to understand the process and to identify the types of sprawl patterns in Bamako. Other purposes of this study are to provide a basis framework for future studies on landscape analysis in the Bamako district, mainly, and secondarily, for the other city landscape environments in the country, and produce perspectives and suggestions that could serve planners and decision-makers.

The main contribution of this paper is that it reveals how to use, analyze, and interpret the retained landscape indexes, and to retrace and identify a long-term urban sprawl pattern over time using remote sensing multi-temporal imagery.

## 2. Materials and Methods

### 2.1. Study Area

Bamako is the capital of the Republic of Mali, located in the southwestern part of the country. Bamako is further subdivided into six municipalities located in southern Mali (Figure 1). Bamako has a maximum area of 267 square kilometers and 3,337,122 inhabitants, with a density of 1115 people per square kilometer (source: Bamako City, 2016). It is located on both sides of the Niger River at 8°0′0″ W and 12°39′0″ N. Bamako is divided into two parts, the north bank and the south bank.

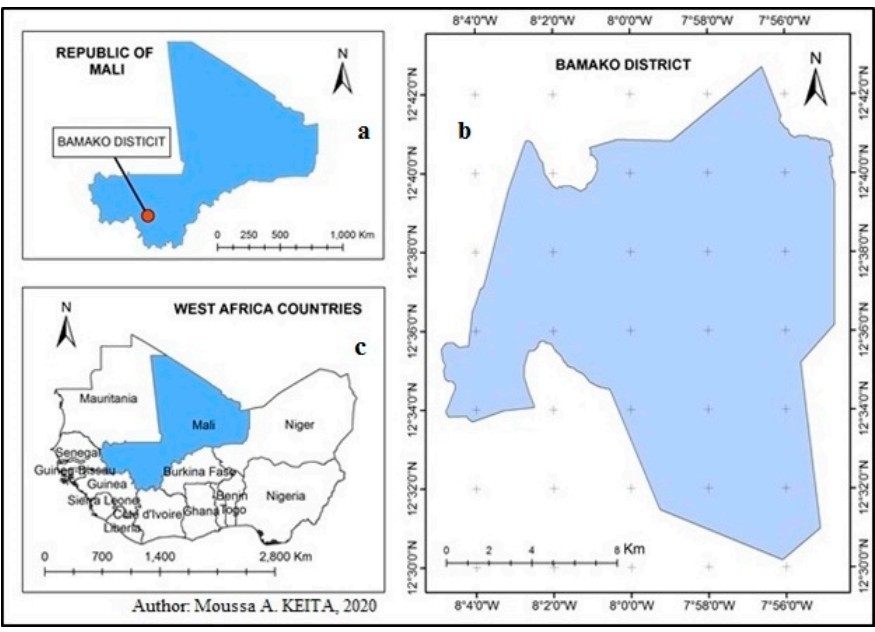

**Figure 1.** Study area location map: (**a**) Bamako district map; (**b**) map of the republic of Mali; (**c**) west Africa map.

For its geological structure and soil, the Bamako district is located in a granite basement covered with sandstone deposits. The river is deeper into the basement of the leaf rock and granite and sedimentary layers. There are two types of surface formations: soil caused by rock change and lateralization, and alluvial formation that occupies the river's primary and secondary riverbeds and their tributaries. The dry season is located in northern Sudan, which starts from November to April, and winter from May to October. The average annual rainfall is between 750 and 1400 mm, with the driest months (January, February) to 290 mm, and the heaviest rainfall (August) at 290 mm. The annual temperature change is 6.7 degrees Celsius. In May, the average temperature is 31.5 degrees Celsius. The vegetation formation of the Bamako area is a gallery forest of savannahs and rivers.

### 2.2. Materials

For the study, using Earth Explorer, four Landsat images for 1990, 2000, 2010, and 2018 were downloaded. To mitigate seasonal effects, which often result in errors in change detection, only images obtained in the summer are used. This avoids the uncertainty of annual variability [22]. Table 1 shows the characteristics and details of these images. The Bamako shape file was used as secondary data acquired from satellite imagery of the study area boundary. Remote sensing software (ENVI), geographic information system software (Arc GIS), and space analysis program software (FRAGSTATS) were used for further data analysis and processing.

**Table 1.** Characteristics of remotely sensed data.

| Landsat | Path | Row | Sensor | Spatial Resolution | Bands Number | Radiometric Resolution | Acquisition Date |
|---------|------|-----|--------|--------------------|--------------|------------------------|------------------|
| Landsat 5 | 199 | 51 | TM | 30 m | 7 | 8 bits | 11 September 1990 |
| Landsat 5 | 199 | 51 | TM | 30 m | 7 | 8 bits | 30 December 2000 |
| Landsat 7 | 199 | 51 | ETM + | 30 m | 7 | 8 bits | 2 December 2010 |
| Landsat 8 | 199 | 51 | OLI-TIRS | 30 m | 11 | 16 bits | 6 January 2018 |

OLI-TIRS: Operation Land Imager-Thermal Infrared Sensor; TM: Thematic Mapper; ETM+: Enhance Thematic Mapper Plus.

### 2.3. Methods

#### 2.3.1. Images Pre-Processing

The radiometric calibration and atmospheric correction were performed after layer stacking. The atmospheric correction benefit is carried out to reduce the atmospheric effects on the electromagnetic radiation. The Fast Line-of-sight Atmospheric Analysis of Hypercubes (FLAASH) module was used as the main tool for atmospheric correction of multispectral and hyperspectral images that operate in the wavelength range from visible to infrared (above 3 μm). After this, the study area boundaries were extracted from the preprocessed images using the Bamako district administrative limits shape file in ArcGIS by the clipping method.

#### 2.3.2. Land Cover Classification

Seven classes were defined based on local conditions of the study area and other papers, using a supervised approach. These classes are: built-up, forest, water, farmland, grassland, bare land, and rock. Class descriptions are shown in Table 2. "The supervised classification is the process of identification of classes within a remote sensing data with inputs from as directed by the user in the form of training data" [37]. The used supervised classification technique is the Maximum Likelihood Classification (MLC) approach. This method of classification calculates the probability for a given pixel to each class and then the pixel will be allocated to a particular class with the highest probability. It calculates the mean and covariance matrix for the training samples and assumes that the pixel values are normally distributed. "Then a probability density function is defined and the input pixels are mapped based on the likelihood that the pixel belongs to that particular class" [38]. The advantage of this sophisticated classifier is that it provides good separation between classes if the training set is strongly and sufficiently defined.

Training samples were selected for each class; supervised MLC algorithm was applied on all images [39]. The number of used training samples per class, according to years, is provided in Table 3. In order to improve the accuracy of the land cover classification results, some spectral indices (NDVI: Normalized Difference Vegetation Index, NDBI: Normalized Difference Built-up Index, and NDWI: Normalized Difference Water Index) were extracted and applied to each classification's result for the second supervised Maximum Likelihood Classification. Finally, 5 × 5 Majority/Minority filter was applied to each land cover classification result, to reduce "salt and pepper" [40].

**Table 2.** Classification class descriptions.

| Class Code | Class Name | Class Description |
| --- | --- | --- |
| 1 | Built-up | Residential areas, settlement areas, industrial zones, commercial zones, facilities, transportation networks. |
| 2 | Forest | Natural vegetation, reserve vegetation areas |
| 3 | Water | River |
| 4 | Farmland | Cereals croplands, vegetables croplands, orchard lands |
| 5 | Grassland | Grasses, shrubs, pasture |
| 6 | Bare land | Non-vegetation and non-cultivate areas, |
| 7 | Rock | Mountain rocks, river rocks, and hill rocks |

**Table 3.** Distribution of reference data samples of the classification accuracy assessment.

| | 1990 | 2000 | 2010 | 2018 |
| --- | --- | --- | --- | --- |
| Built-up | 150 | 250 | 300 | 350 |
| Vegetation | 70 | 70 | 50 | 50 |
| water | 50 | 50 | 50 | 50 |
| Farmland | 150 | 150 | 100 | 50 |
| Grassland | 50 | 50 | 50 | 50 |
| Bare land | 60 | 60 | 50 | 50 |
| rock | 50 | 50 | 50 | 50 |
| **Total** | **580** | **680** | **650** | **650** |

Note that, the linear features, such as road classes, were drawn manually based only on visual interpretation, because of the low-resolution of the images.

### 2.3.3. Post-Classification

Land cover classification results, originally in raster format, were converted into shape file vector format in ArcGIS 10.2 for LC mapping. Four LC maps were produced for each selected year (1990, 2000, 2010, and 2018).

In order to verify the reliability of the land use/cover classifications results, accuracy assessment was performed in ENVI using the initial sample of the supervised classification as reference data, and confusion matrix for each land cover classification result was created. A total of 590 reference data were used for the result of 1990, 690 for 2000, and 650 for 2010 and 2018. The distribution of reference data between classes is presented is presented in Table 3. The relative increased or decreased numbers of reference data are linked to the changes of class area over time. Descriptive and analytical statistics, such as producer's accuracy, user's accuracy, and overall accuracy, were derived from each classification error matrix [41,42].

### 2.3.4. Landscape Pattern Changes Detection

In this study, change detection and analysis mainly focused on landscape pattern analysis at both landscape and class levels in Bamako, and between communes and river sides for comparative analysis. Class indexes were used to represent the spatial distribution and pattern within a landscape of a single patch type; landscape indices represented the spatial pattern of the entire landscape mosaic, considering all patch types simultaneously [43]. For this purpose, a

number of suitable indexes were selected to describe landscape composition and configuration over the time. Features, such as patch shape, size, quantity, and spatial combination [16–44], were incorporated to define the complexities and arrangements of the landscape pattern by indices. The landscape pattern analysis program software package FRAGSTATS (version 4.2) was used to calculate the selected landscape metrics, both at landscape and class level from categorical landscape maps. Then, categorical landscape maps were focused on land cover time series maps, because of familiarity to managers, long history of use in landscape ecology, and the fact that land management agencies largely base planning and analysis on such kind of representation of landscape structure [45]. The spatial resolution of 30 m for all categorical maps was used in this study. The description of the selected indices for class and landscape level used in the current study is given in Tables 4 and 5, respectively. The methodology used for landscape pattern analysis is presented in Figure 2.

**Table 4.** Class metrics and descriptions [43–46].

| Class Metric | Formula | Description |
|---|---|---|
| Percentage of landscape (%PLAND) | $p_i = \frac{\sum_{j=1}^{n} a_{ij}}{A}(100)$ | %LAND equals the sum of the areas (m$^2$) of all patches of the corresponding patch type, divided by total landscape area (m$^2$), multiplied by 100 (to convert to a percentage). |
| Class Area (CA) | $CA = \sum_{n=1}^{\infty} a_{ij}\left(\frac{1}{10000}\right)$ | CA equals the sum of the areas (m2) of all patches of the corresponding patch type, divided by 10,000 (to convert to hectares); which is total class area. CA approaches 0 as the patch type becomes increasing rare in the landscape. |
| Number of Patches (NP) | $NP = n_i$ | Number of patches of corresponding patch type (class). |
| Largest Patch Index (LPI) | $LPI = \frac{\max\left(a_{ij}\right)_{j=1}^{n}}{A}(100)$ | LPI equals the area (m$^2$) of the largest patch of the corresponding patch type divided by total landscape area (m$^2$), multiplied by 100 (to convert to a percentage). |
| Landscape Shape Index (LSI) | $LSI = \frac{0.25 \sum_{k=1}^{m} e_{ik}}{\sqrt{A}}$ | LSI equals the sum of the landscape boundary and all edge segments (m) within the landscape boundary involving the corresponding patch type (including those bordering background), divided by the square root of the total landscape area (m$^2$). |

**Table 5.** Landscape Metrics and descriptions adopted from [43–47].

| Landscape Metric | Formula | Descriptions |
|---|---|---|
| Number of Patches (NP) | $NP = N$ | NP equals the number of patches in the landscape. NP does not include any background patches within the landscape or patches in the landscape border. |
| Largest Patch Index (LPI) | $LPI = \frac{\max\left(a_{ij}\right)_{j=1}^{n}}{A}(100)$ | LPI equals the area (m$^2$) of the largest patch in the landscape divided by total landscape area (m$^2$), multiplied by 100 (to convert to a percentage); in other words, LPI equals the percentage of the landscape that the largest patch comprises. |
| Landscape Shape Index (LSI) | $LSI = \frac{0.25E'}{\sqrt{A}}$ | LSI equals the sum of the landscape boundary (regardless of whether it represents true edge) and all edge segments (m) within the landscape boundary (including those bordering background), divided by the square root of the total landscape area (m$^2$), and adjusted by a constant for a circular standard (vector) or square standard (raster). |
| Contagion (CONTAG) | $CONTAG = \left[1 + \sum_{i=1}^{m} \sum_{j=1}^{m} \frac{p_{ij}\ln\left(p_{ij}\right)}{2\ln(m)}\right](100)$ | CONTAG equals 1 plus the sum of the proportional abundance of each patch type multiplied by number of adjacencies between cells of that patch type and all other patch types, multiplied by the logarithm of the same quantity, summed over each patch type; divided by 2 times the logarithm of the number of patch types; multiplied by 100 (to convert to a percentage). |

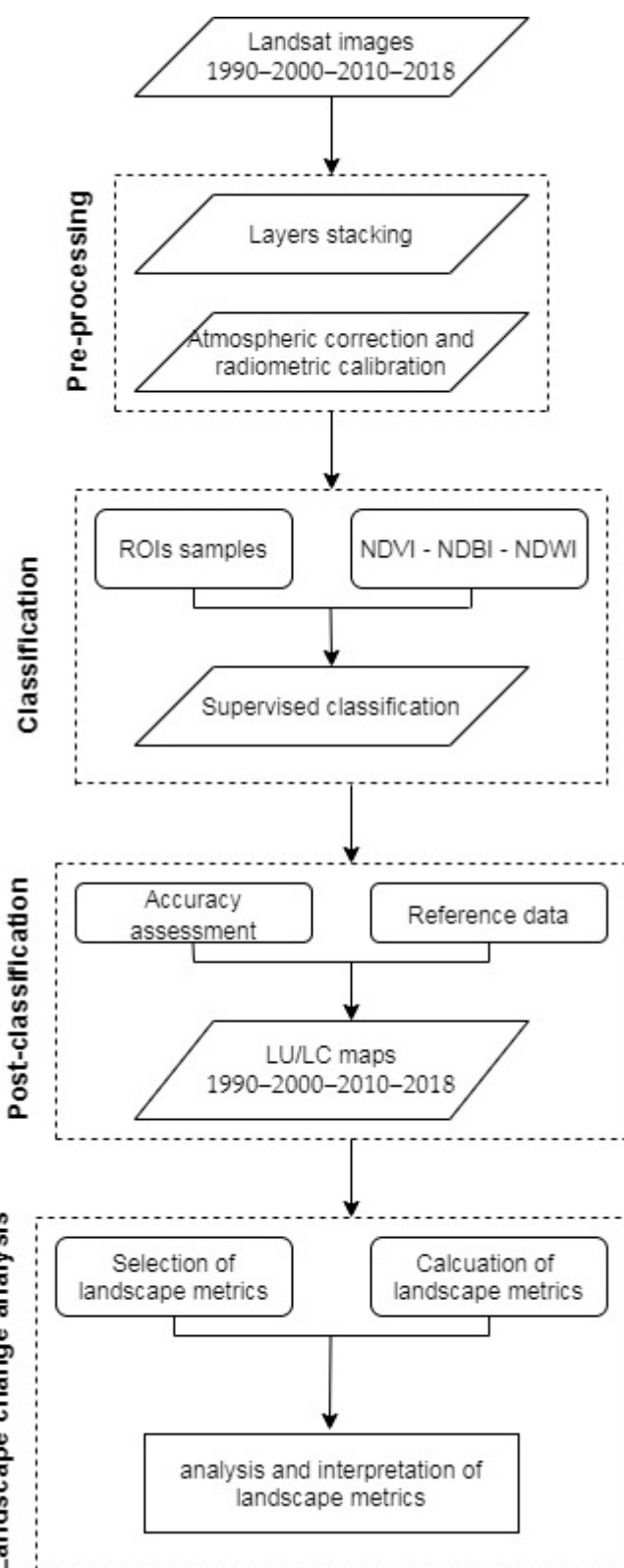

**Figure 2.** Methodology flowchart.

## 3. Results and Discussions

### 3.1. Land Cover/Land Use Maps

In this study, four LC thematic maps were produced by applying the supervised classification method on multitemporal Landsat images. Figure 3 illustrates the classification

results. There is a total of eleven classes; however, the classification has been applied on only those classes (7) that represent area feature objects (built-up, forest, water, farmland, grassland, bare land, and rock). Linear feature classes, such as roads (main road, secondary road, tertiary road, and trunk), have been delineated as shape files using visual interpretation. Thus, information produced from road features do not reflect real information of the entire road features of the real world, but give useful information on the spatiotemporal evolution of roads with an account of the variation of length, as shown in Table 6. A study on the important changes among road features from 1990 to 2018 showed that there was no change in the trunk class after 1990 (24 km for all the years), while other roads showed a visible increase, and continuous change over time, with a considerable change between 2010 and 2018 for all classes.

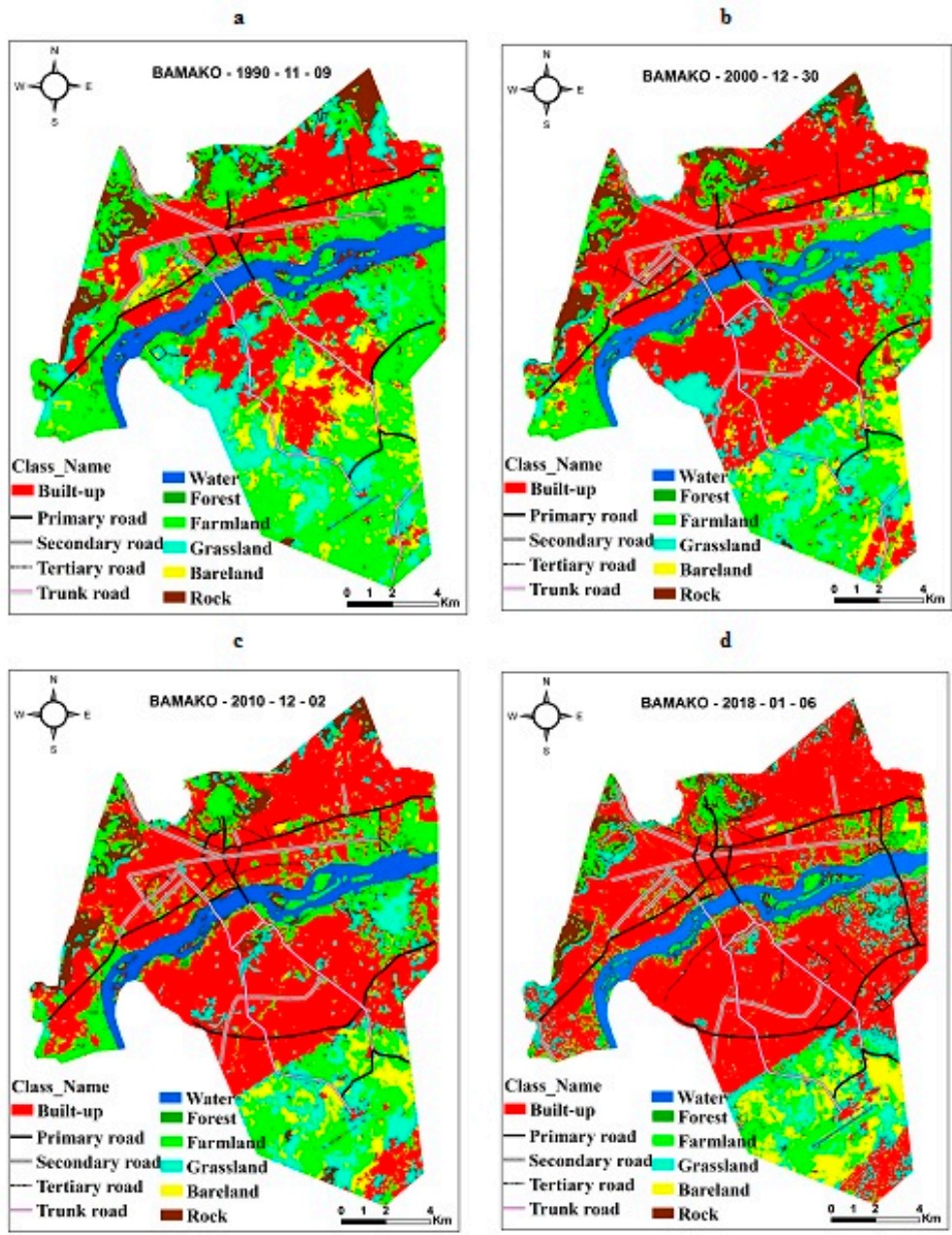

**Figure 3.** Land cover classification maps in Bamako district from 1990 to 2018. (**a**) land cover map 1990; (**b**) land cover map 2000; (**c**) land cover map: 2010; (**d**) land cover map 2018.

**Table 6.** Roads statistics per kilometer from 1990 to 2018.

| Class Name | 1990 | 2000 | 2010 | 2018 |
|---|---|---|---|---|
| Primary road | 37 | 37 | 47 | 55 |
| Secondary road | 25 | 38 | 39 | 48 |
| Tertiary road | 9 | 14 | 16 | 32 |
| trunk | 15 | 24 | 24 | 24 |

According to classification results, class area, and growth ratio statistics (as shown in Table 7), remarkable changes occurred in the study area from 1990 to 2018. These changes were more significant in built-up and farmland classes. Over time, the built-up area was increased from 6546 ha (in 1990) to 14,774 ha (in 2018), while, farmland reduced from 8841 ha (in 1990) to 3141 ha (in 1990). With this shift, the land use built-up area emerged as the most dominant land cover type; and the major land cover changes within the study area are concerned with both built-up and farmland. Growth ratio confirms that the major land cover changes have occurred on built-up and farmland within the study area. The positive ratio values were observed in built-up, with the highest ratio value (18.18%) in 2018. All other classes have a negative growth ratio values, except bare land, with a ratio of 6.92 in 2000. The highest negative growth ratio was observed for farmland (−25.09) in 2018, which implies that farmlands are being converted to other land uses, especially build-up in Bamako, at a very fast rate.

**Table 7.** Class areas and growth ratio (using 1990 as baseline) statistics per hectare from 1990 to 2018.

| Class Name | 1990 | 2000 | | 2010 | | 2018 | | Total |
|---|---|---|---|---|---|---|---|---|
| | Area/ha | Area/ha | Ratio% | Area/ha | Ratio% | Area/ha | Ratio% | Area/ha |
| Built-up | 6546 | 11,606 | 11.18 | 12,308 | 12.73 | 14,774 | 18.18 | 45,234 |
| Forest | 298 | 180 | −13.33 | 142 | −17.62 | 265 | −3.72 | 885 |
| Farmland | 8841 | 5563 | −14.43 | 5168 | −16.17 | 3141 | −25.09 | 22,713 |
| Water | 1557 | 1295 | −4.77 | 1397 | −2.91 | 1243 | −5.71 | 5492 |
| Grassland | 3288 | 2772 | −4.45 | 2529 | −6.56 | 2981 | −2.65 | 11,570 |
| Bare land | 1955 | 2457 | 6.92 | 1253 | −9.66 | 1581 | −5.16 | 7246 |
| Rock | 2013 | 1813 | −3.00 | 1580 | −2.45 | 1239 | 11.64 | 6645 |

LC classification results also showed urban sprawl or growth outside the Bamako district, in the rural surrounding municipalities. The east and the west part of LC maps show sprawl outside the Bamako district with more evidence. Figure 4 also shows that urban sprawl is more rapid these days in the surrounding municipalities of Bamako than the district expansion.

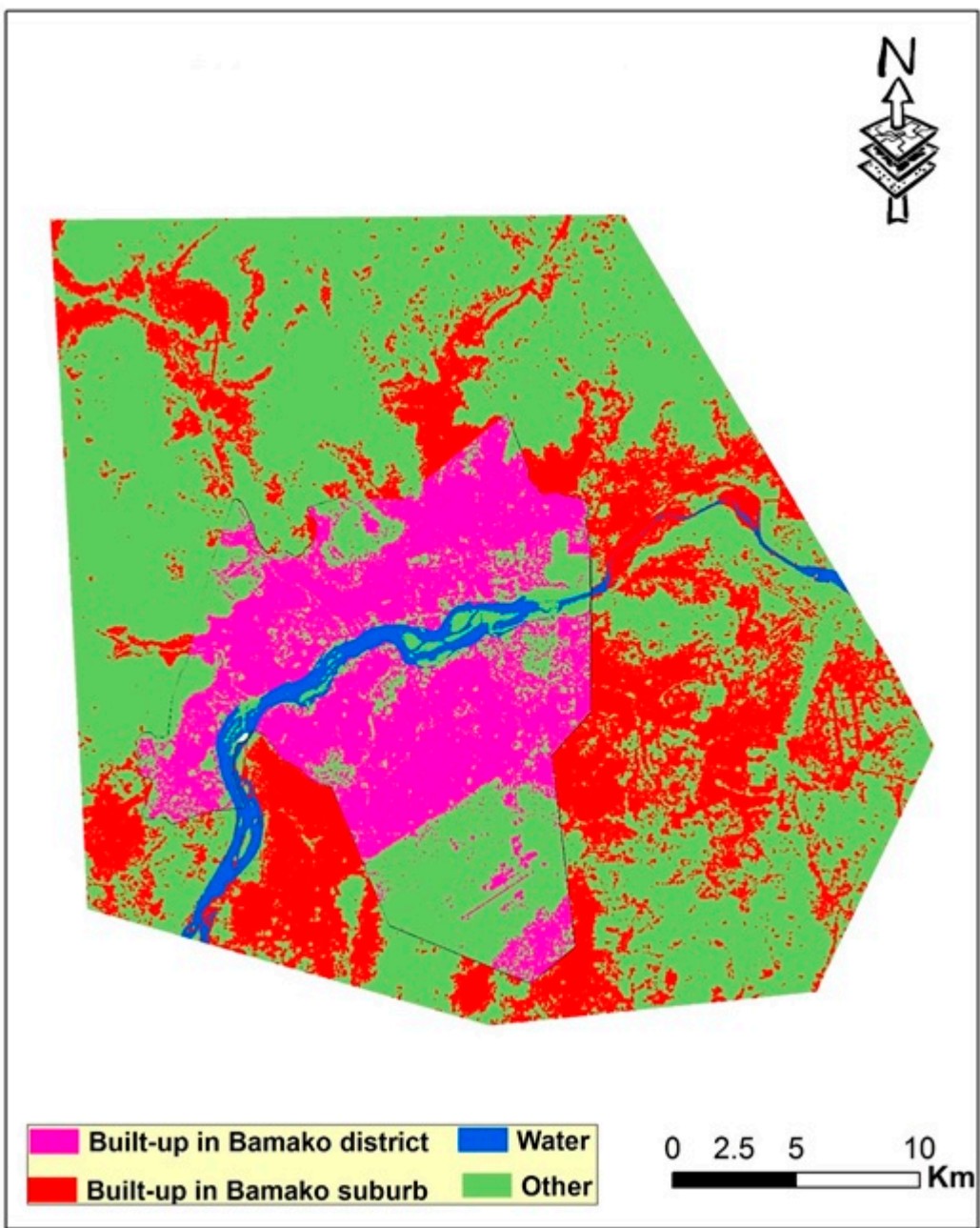

**Figure 4.** Urban sprawl status in Bamako district and its suburbs areas in 2018.

### 3.2. Accuracy Assessment

Accuracy assessment was performed to evaluate the reliability of thematic map classification results of four LC maps. The same regions of interest used for each LC classification as training samples were used as reference data to perform accuracy assessment for each thematic map. Overall accuracy values are 93.80, 91.77, 97.35, and 98.94%, respectively for 1990, 2000, 2010, and 2018. Water class is the most accurate followed by built-up class. The overall accuracy values supported the high reliability of the classification results. More details on accuracy assessment results are presented in Table 8.

**Table 8.** Accuracy assessment results of classification.

| Class Code | Class | Producer's Accuracy % | | | | User's Accuracy % | | | |
|---|---|---|---|---|---|---|---|---|---|
| | | **1990** | **2000** | **2010** | **2018** | **1990** | **2000** | **2010** | **2018** |
| 1 | Built-up | 91.89 | 90.99 | 97.79 | 99.27 | 96.68 | 99.26 | 99.80 | 99.80 |
| 2 | Forest | 79.17 | 98.46 | 80.64 | 96.63 | 98.43 | 57.65 | 94.33 | 94.45 |
| 3 | Water | 95.91 | 99.20 | 97.84 | 99.79 | 99.80 | 100 | 100 | 100 |
| 4 | Farmland | 95.66 | 91.60 | 98.44 | 96.17 | 95.39 | 92.47 | 94.21 | 93.06 |
| 5 | Grassland | 91.82 | 83.76 | 91.65 | 98.74 | 87.77 | 61.00 | 94.01 | 98.23 |
| 6 | Bare land | 95 | 98.61 | 97.01 | 99.11 | 95.41 | 90.30 | 72.76 | 94.91 |
| 7 | Rock | 94.93 | 89.89 | 99.68 | 98.64 | 75.74 | 53.69 | 83.42 | 99.39 |
| | | Overall accuracy% | | | | Kappa coefficient | | | |
| **1990** | **2000** | **2010** | **2018** | | **1990** | | **2018** | **2010** | **2018** |
| 93.80 | 91.77 | 97.35 | 98.94 | | 0.91 | | 0.84 | 0.97 | 0.98 |

### 3.3. Landscape Metrics

#### 3.3.1. Landscape Metrics at Class and Landscape Level in the Entire Area of the Bamako District

This study presents four landscape metric results at class level and four at landscape level, over a scale of eighteen years, which is further subdivided into four years (1990, 2000, 2010, and 2018). The trends of these metrics are presented in Tables 9 and 10.

**Table 9.** Landscape metrics at class level in 1990, 2000, 2010, and 2018.

| Classes | Years | PLAND | NP | LPI | LSI |
|---|---|---|---|---|---|
| Built-up | 1990 | 27.1845 | 311 | 13.7012 | 16.5211 |
| | 2000 | 46.1413 | 391 | 21.7469 | 17.1606 |
| | 2010 | 50.1609 | 269 | 21.5993 | 16.9946 |
| | 2018 | 60.0585 | 996 | 55.4013 | 29.3123 |
| Forest | 1990 | 1.2107 | 132 | 0.1837 | 12.1043 |
| | 2000 | 0.732 | 136 | 0.0671 | 12.0778 |
| | 2010 | 0.5786 | 122 | 0.0525 | 12.025 |
| | 2018 | 1.0721 | 387 | 0.0756 | 20.156 |
| Water | 1990 | 6.4418 | 5 | 6.4235 | 6.2755 |
| | 2000 | 5.2467 | 10 | 4.9997 | 7.4042 |
| | 2010 | 5.6894 | 13 | 5.3573 | 6.9639 |
| | 2018 | 5.2023 | 19 | 5.064 | 8.272 |
| Farmland | 1990 | 35.9632 | 388 | 14.134 | 25.5479 |
| | 2000 | 22.6274 | 485 | 2.8391 | 27.9014 |
| | 2010 | 21.0479 | 488 | 2.6065 | 27.4551 |
| | 2018 | 13.1367 | 1169 | 1.4829 | 41.3799 |
| Grassland | 1990 | 13.5781 | 387 | 4.9379 | 23.3818 |
| | 2000 | 11.2811 | 498 | 2.8578 | 26.0285 |
| | 2010 | 10.2098 | 491 | 1.3706 | 26.8234 |
| | 2018 | 11.2016 | 1458 | 1.0053 | 44.9886 |
| Bare land | 1990 | 8.1192 | 452 | 1.6674 | 24.6779 |
| | 2000 | 7.3376 | 641 | 0.6494 | 26.3993 |
| | 2010 | 5.1082 | 345 | 0.7149 | 19.5508 |
| | 2018 | 6.4376 | 329 | 1.3376 | 18.6906 |
| Rock | 1990 | 7.5026 | 481 | 1.2665 | 23.1608 |
| | 2000 | 6.6339 | 356 | 0.8413 | 21.2528 |
| | 2010 | 7.2052 | 365 | 1.1921 | 21.3772 |
| | 2018 | 2.8913 | 302 | 0.7743 | 18.5955 |

PLAND is percent of landscape; NP is number of patches; LPI is largest patch index; and LSI is landscape shape index.

**Table 10.** Landscape metrics in landscape level in 1990, 2000, 2010 and 2018.

| Years | NP | LPI | LSI | CONTAG |
|-------|------|---------|---------|---------|
| 1990 | 2156 | 14.134 | 25.8132 | 47.7392 |
| 2000 | 2517 | 21.7469 | 25.5062 | 50.7749 |
| 2010 | 2093 | 21.5993 | 24.3075 | 53.1526 |
| 2018 | 4660 | 55.4013 | 33.2837 | 54.1367 |

At the class level, statistics of the percentage of landscape (%PLAND) show that the built-up area was the dominant class with increase percentages of 27.18, 46.14, 50.16, and 60.05%, respectively for 1990, 2000, 2010, and 2018. Inversely, %PLAND of farmland class decreased over the years at the rate of 35.96, 22.62, 21.04, and 13.13%, respectively, for 1990, 2000, 2010, and 2018. There were no significant changes in %PLAND for the other class types, except rock class in 2018 (with 2.89%) and 7.502% in 1990, because, due to the lack of space for construction in recent years, urban sprawl is undergoing in the mountain areas.

The statistics of the number of patches (NP) show that classes in the landscape were becoming more fragmentized over the years, especially in the case of grassland with NP of 387 in 1990, 498 in 2000, 491 in 2010 and 1458 in 2018. The second fragmented class is the farmland followed by rock (481NP), bare land (452 NP), farmland (388NP) in 1990; bare land (641NP), grassland (498NP), and farmland (485NP) in 2000; grassland (491NP), farmland (488NP), and rock (365NP) 2010; grassland (1458NP), farmland (1169NP), and built-up (996NP) in 2018. NP value for a given class is high when this class type is fragmented more in the landscape.

For the largest patch index (LPI), the highest values were observed in the built-up class, and the lowest values in the forest class; and these values increased over the years. For built-up, these values were 13.70% in 1990, 21.74% in 2000, 21.59% in 2010, and 55.40% in 2018. It implies that the largest index in 1990 is lesser than the largest one in 2018 and vice versa. LPI approaches 0 when the major patch of the parallel patch type becomes progressively smaller. LPI is 100 when the whole landscape consists of a single patch of the parallel patch type [25]. In the other landscape, there is no significant change in the LPI values; it changes according to the year. Based on LPI values, it can be inferred that the landscape shapes of each corresponding patch type in the landscape are more irregular.

Based on the LSI values, the most irregular patches were observed in 2018 for built-up (29.31), forest (20.15), water (8.27), farmland (41.37), and grassland (44.98). In 2000, a similar pattern was observed for bare land (26.39), while in 2010, it was for rock (21.3772). It can be argued that the highest length of edge within the landscape of these corresponding patch types is in the same years. LSI = 1 when the landscape consists of a single patch of the consistent type and is circular (vector) or square (raster); LSI upsurges without limit as the landscape form becomes more uneven, or as the length of edge inside the landscape of the consistent patch type upsurges (or both) [43].

In regards to built-up indices, the trends of NP, LPI, and LSI showed an increasing fragmented development of built-up patterns, except from 2000 to 2010, where NP decreased slightly. That is to say that the built-up pattern underwent scattered development during the study period, except from 2000 to 2010, where infill development of the built-pattern was undergoing. Moreover, analysis of built-up pattern changes based on produced LC map analysis show some ribbon development of the built-up pattern along major roads, rivers, mountains, and valleys.

In general, the study shows an increase in the value of indices between 1990 and 2000, a slight decrease between 2000 and 2010, and then a considerable increase from 2010 to 2018. Forest class covers the smallest area in the study area and is the weakest class in terms of NP and LPI. It is inferred that forest is the less fragmented class and has the smallest patch with the landscape. The grassland was designated as a class with the highest fragmentation, farmland with the highest shape irregularity, and built-up with the highest patches. The detailed general trends of theses landscape metrics are given in Table 9 and Figure 5.

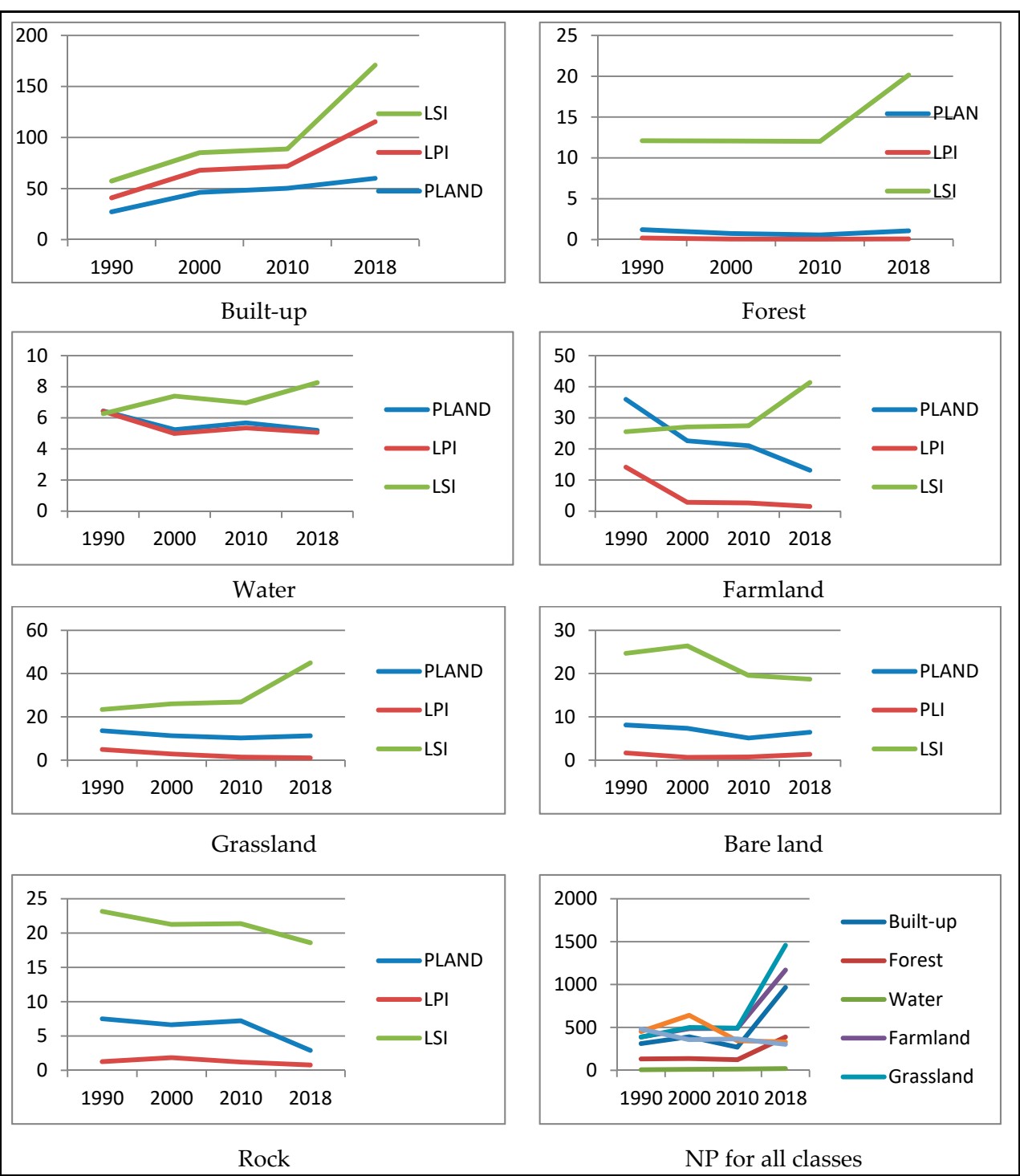

**Figure 5.** Curves of index trends at class level.

At the landscape level, the highest number of patches in the entire landscape is located in 2018 (4660 NP), with the lowest in 2010 (2093 NP). In other words, the landscape is more fragmented and heterogeneous in 2018 than 2010. The second more fragmented and heterogeneous landscape occurred in 2000 (2517 NP). The rapid increase in the NP in the landscape throughout the study period could be explained by sustained urban sprawl. This could be a signal of varied and fragmented sprawl development. As for largest patch index, the highest value was observed in 2018, occupying 55.40% of the entire landscape, and the lowest was in 1990, with 14.13%. LPI approaches 0 when the largest patch in the

landscape is progressively small. LPI reaches 100 when the whole landscape contains a solitary patch—that is, when the major patch encompasses 100 percent of the landscape [43]. Based on the values of LSI, it can be said that landscape shapes were irregular within each landscape for all years. The most irregular landscape shape was observed in 2018 with a value of 33.28. The irregularity of shapes within the landscape for all the years inferred that there was no significant change of shape lengths over the years, except for 2018. LSI is 1 when the landscape contains a single circular (vector) or square (raster) patch. LSI increases without limit as the landscape shape becomes more uneven, or as the length of the edge inside the landscape upsurges (or both) [43]. For the landscape contagion index (CONTAG), the study notices a mean distribution of the patches types adjacencies within the landscape. The CONTAG indexes values varies from 47.73% in 1990, 50.77% in 2000, and 53.15% in 2010, to 54.13% in 2018. That means a slight increase the distributions of the adjacency and heterogeneity of landscape patches over the years. CONTAG is close to 0 as the distribution of adjacent (single cell level) between unique patch types becomes less and less uniform. CONTAG is 100 when all patch types are equal to all other patch types [43].

Overall, an increase for all indexes over the years was noticed, except in 2010 for NP, LPI, and LSI (Table 10 and Figure 6).

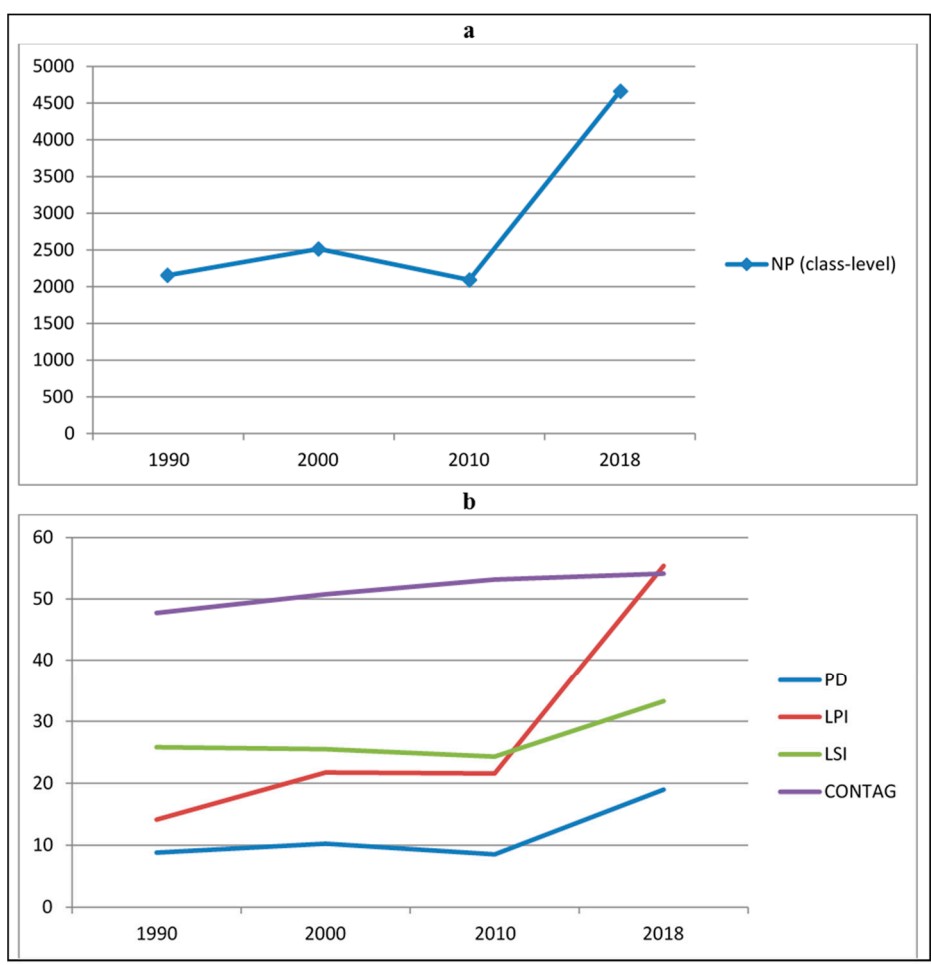

**Figure 6.** Curves trends of indexes at landscape level.

### 3.3.2. Landscape Metrics and Built-Up Pattern Change between Communes

According to the class area (CA) values from built-up class, it has been remarked that, over the study period, commune 6 registered the highest value of built-up area followed by commune 5, except in 1990, where commune 1 scored the second highest area of built-up. Then, in 2000, communes 5 and 4 became the second and third communes, respectively, after commune 6, in terms of the highest built-up areas. Communes 3 and

2 scored, respectively, the lowest values of built-up areas over the study period. This is normal because these communes are located in the city center and are enclosed and blocked in their sprawl, on the one hand, by mountains and rivers, and on the other hand, by communes 1 and 4. Figure 7 shows the trends of CA curves.

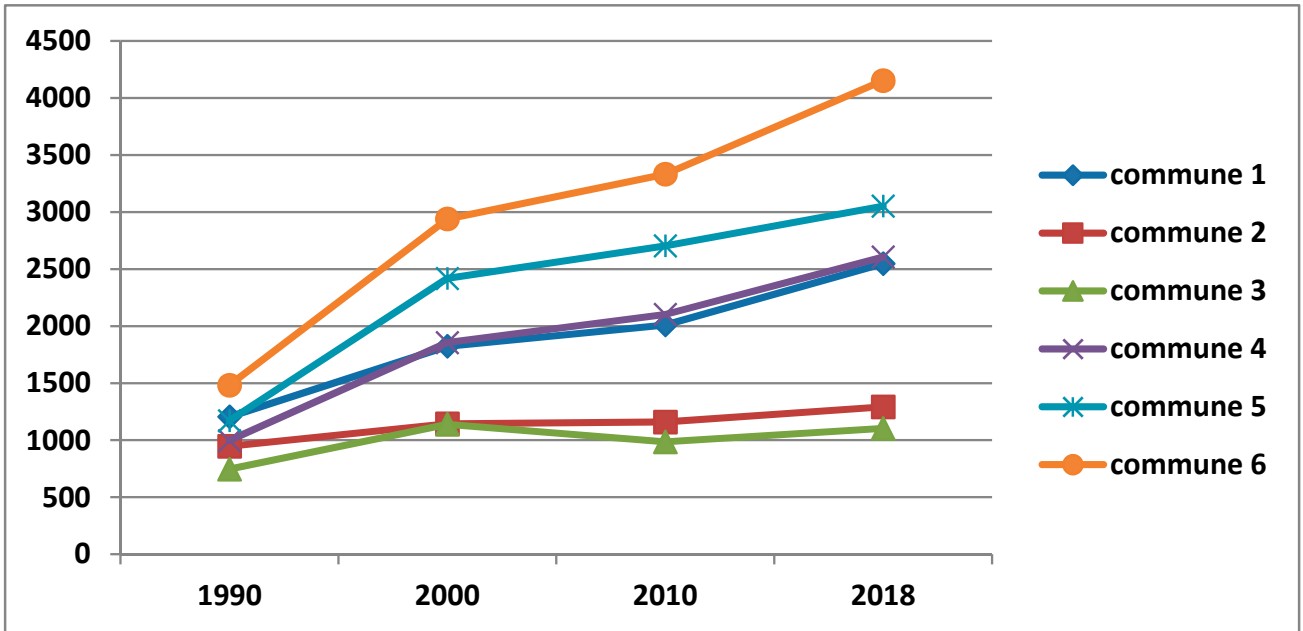

**Figure 7.** Curve trends of CA between communes in Bamako district.

PLAND values increase for each commune over the study period; however, it is important to note that there is no agreement between trends of CA and PLAND by making comparisons between communes. Communes 3 and 2, which registered the lowest values of CA, have almost the highest value of PLAND over the study period, while commune 6 which scored the highest value of CA, had the lowest value of PLAND in the study period. This is not normal, but it can be explained by the effects of the difference of spatial domain, as mentioned before. Communes 3 and 2 have the smallest areas among the communes. A smaller space domain may have a percentage of PLAND more than a larger space domain, so the class considered is larger, in a larger space domain than in the smaller one.

Concerning the configuration of the built-up class, trends of NP values show that the built-up class in each commune has undergone a fragmented development process of sprawl over the study period except in 2010, where all values of NP decreased, except for commune 6, which remained the most fragmented commune. Communes 6 and 4 developed the highest level of fragmented sprawl process. The decrease trend in NP values could be explained by an infill development of sprawl, which fill the open spaces between built-up, and merge smaller patches to bigger patches, creating a decrease in NP. Figure 8 shows the trends of the NP curve.

The highest values of the largest patch (LPI) are registered in 1990 and 2000 in commune 2 (50.39 and 61.10, respectively), in 2010 and 2018 in commune 5 (67.50 and 75.29, respectively), followed by commune 2, while the lowest values of LPI are registered, respectively, during the study period in commune 6 and 4, which registered the more fragmented built-up sprawl process over the study period. Thus, it could be concluded that, the higher the LPI value, the more the landscape is compact and less fragmented, and when the value of LPI is low, the landscape is less compact and more fragmented. For example, a value of LPI of 50.39 means that the largest patch of built-up class occupies an area of 50.39% within the total area of built-up class (or built-up landscape).

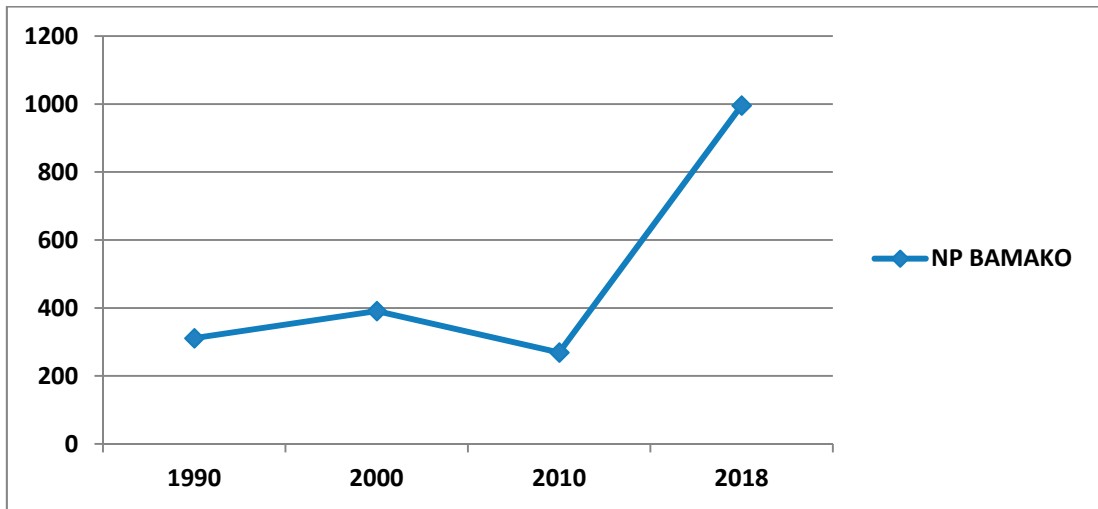

**Figure 8.** Curve of NP for built-up in Bamako district from 1990 to 2018.

Trends of LSI values also confirmed that commune 6 and 4 had more irregular shapes, were less compact, and more fragmented because they had the highest value of LSI. When LSI equals 1, the landscape consists of a single patch; LSI increases without limit as the landscape shape becomes more irregular or as the length of the edge within the landscape increases (or both) [25]. Table 11 shows more details regarding trends of metrics, and Figures 7 and 8 show the curve trends of landscapes metrics.

**Table 11.** Landscape metrics at class level for built-up between communes in Bamako.

| Commune | CA | PLAND | NP | LPI | LSI |
|---------|------|-------|------|------|------|
| **1990** | | | | | |
| Commune 1 | 1205.19 | 34.26 | 38 | 32.94 | 5.71 |
| Commune 2 | 946.80 | 51.30 | 19 | 50.39 | 4.80 |
| Commune 3 | 746.10 | 37.53 | 41 | 33.92 | 6.07 |
| Commune 4 | 996.75 | 22.23 | 59 | 11.52 | 7.66 |
| Commune 5 | 1170.27 | 29.47 | 44 | 16.38 | 6.88 |
| Commune 6 | 1482.12 | 17.04 | 133 | 9.65 | 10.26 |
| **2000** | | | | | |
| Commune 1 | 1823.04 | 51.78 | 64 | 49.36 | 6.26 |
| Commune 2 | 1143.45 | 61.79 | 11 | 61.10 | 4.52 |
| Commune 3 | 1138.95 | 57.31 | 36 | 50.38 | 6.28 |
| Commune 4 | 1855.98 | 41.34 | 91 | 31.44 | 8.40 |
| Commune 5 | 2418.39 | 60.77 | 36 | 59.73 | 6.58 |
| Commune 6 | 2939.31 | 33.74 | 187 | 25.39 | 11.64 |

**Table 11.** *Cont.*

| 2010 | | | | | |
|---|---|---|---|---|---|
| Commune | CA | PLAND | NP | LPI | LSI |
| Commune 1 | 2009.97 | 57.15 | 43 | 53.73 | 6.82 |
| Commune 2 | 1158.57 | 62.88 | 16 | 60.90 | 5.37 |
| Commune 3 | 985.05 | 49.59 | 30 | 40.52 | 6.53 |
| Commune 4 | 2102.40 | 46.89 | 51 | 31.69 | 8.06 |
| Commune 5 | 2703.24 | 67.97 | 23 | 67.50 | 5.49 |
| Commune 6 | 3331.51 | 38.29 | 136 | 29.86 | 11.39 |
| **2018** | | | | | |
| Commune | CA | PLAND | NP | LPI | LSI |
| Commune 1 | 2548.08 | 72.34 | 86 | 70.56 | 10.93 |
| Commune 2 | 1291.50 | 69.78 | 67 | 64.92 | 7.96 |
| Commune 3 | 1103.31 | 54.41 | 127 | 49.75 | 11.52 |
| Commune 4 | 2605.59 | 57.97 | 257 | 54.25 | 15.68 |
| Commune 5 | 3052.08 | 76.65 | 121 | 75.29 | 7.60 |
| Commune 6 | 4152.51 | 47.68 | 372 | 39.54 | 19.21 |

Thematic maps from classification used to extract indexes for each commune and each river side are presented in the index as Supplementary Materials.

### 3.3.3. Landscape Metrics and Built-Up Pattern Change between River Sides

Trends of CA and PLAND values show that the areas of built-up classes were more important on the river's left side than the right side throughout the study period. CA varied on the river's left side (RLS) from 1990 to 2018, with values ranging from 3894.03 to 7556.04 ha, an increase of 3662.01 ha (7556.04–3894.03); while it varied during the same period on the river's right side (RRS), with values ranging from 2651.31 to 7207.92 ha, an increase of 4556.61 ha (7207.92–2651.31). It is important to note that, despite the CA and PLAND values being higher on the RLS than the RRS, the gain in built-up area over the study period is more important on the RRS than the RLS so it could be concludes now that sprawl, or changes in built-up patterns, were more important on the RRS than on the RLS

Trends of NP values are high on the RRS than on the RLS, except in 2018, which means that built-up on the RRS was more fragmented than the one on the RLS. The lowest degree of fragmented development on the two sides is registered in 1990, and the highest fragmented development is registered in 2018; so built-up pattern was more fragmented on each side of the river.

NP value trends decreased on both sides, from 2000 to 2010, which could be explained by an infill development period of sprawl. LPI values show that the largest patch was larger on the RLS than on the RRS, and LSI values show that built-up pattern was more irregular on the RRS than on the RLS; in other words, the built-up shapes on the RRS were less compact than the one on the RLS.

Despite some difference in metric values, both sides developed the same trends of development over time, as curves show trends in Table 12.

**Table 12.** Landscape metrics for built-up between the river's left side and right side.

| Year | PLAND | | NP | | LPI | | LSI | | CA | |
|---|---|---|---|---|---|---|---|---|---|---|
| | RLS | RRS | RLS | RRS | RLS | RRS | RLS | RRS | RLS | RRS |
| 1990 | 32.91 | 20.94 | 142 | 169 | 27.74 | 9.94 | 11.18 | 12.02 | 3894.03 | 2651.31 |
| 2000 | 50.33 | 42.22 | 187 | 214 | 45.01 | 36.19 | 11.69 | 12.68 | 5961.15 | 5355.45 |
| 2010 | 52.92 | 47.63 | 118 | 151 | 44.71 | 41.69 | 12.32 | 11.80 | 6258.24 | 6035.67 |
| 2018 | 63.74 | 56.82 | 504 | 493 | 61.00 | 50.99 | 22.38 | 19.42 | 7556.04 | 7207.92 |

Note: RLS: river left side; RRS: river right side.

The findings demonstrate the utility of remote sensing imageries and landscape metrics for analysis of urban environments. The results also provide useful information and understanding about the study area environment and dynamic changes over the study period. The multiscale analysis done in this study makes comparisons between communes, in terms of built-up or sprawl pattern changes. Thus, this method identifies the degree of urban sprawl in each commune and each river side. This study also reveals and shows how to use, analyze, and interpret the retained landscape indexes to retrace and identify long-term urban sprawl patterns over time, by using remote sensing multi-temporal imagery.

However, it could be possible to create more details in class types, but the low-resolution of the images did not allow that. This resolution problem did not allow to directly integrate the road entities in the supervised classification as mentioned above. High-resolution of images is required to produce deep details by increasing or separating more features for classes in the landscape identify within built-up: commercial and industrial area, settlement area, residential area, administrative area, etc. Data acquired from the very high-resolution sensor might be helpful in deriving more detailed classification of the landscapes [48–50]. It will be helpful in developing a deep understanding of landscape composition and configuration of the desired study area, mainly in developing countries, where urban structures and construction patterns are usually of smaller scale, and higher incomplexity, than in developed countries [51]. However, the pertinence of the findings of this research is still unaffected by this gap. It can be considered as the basis of such research and developing future plans for Bamako city management.

This study utilized the landscape metrics method in the Bamako district, with metrics that were successfully used in other places, in order to monitor landscape changes based on index analyses [5,35].

The main importance of this test study is that it could be a basis or framework for future studies on landscape analysis in the Bamako district (mainly) and secondarily for other cities in the country. It could also serve planners or decision-makers when implementing suitable policies and reliable management mechanisms on urban sprawl.

**4. Conclusions**

For this study, four different land cover maps from Landsat images of 1990, 2000, 2010, and 2018 were used to evaluate a set of four selected metrics at class level, and five selected metrics at landscape level to reveal patterns and changes of urban sprawl in the study area. The findings prove a major dynamic in the landscape throughout the study, with major changes in built-up and farmland. The LC maps, area, and growth ratio statistics from classification could be helpful in the visualization of the real change in landscape within the study area. Index statistics, in terms of PLAND, NP, LPI, and CONTAG also showed significant changes. Based on index statistics at class level, the study concluded that built up gained a significant area in terms of size (with a PLAND from 27.18% in 1990 to 60.05% in 2018). Farmland lost the maximum area (with a PLAND from 35.96% in 1990 to 13.13% in 2018). The most fragmented class was bare land (387 NP in 1990 and 1458% NP in 2018) and the less fragmented was water (5 NP in 1990 and 19 NP in 2018). Grassland suffered maximum fragmentation, farmland with the highest shape irregularity, and built-up with the highest patches. The findings also noticed that, at the landscape level, the highest number of patches in the entire landscape occurred in 2018 (4660 NP), and the lowest in

2010 (2093 NP). In other words, the landscape was more fragmented in 2018 than 2010. The second, more fragmented landscape, was observed in 2000 (2517 NP). Urban sprawl was more important in communes 6, 5, and 4, respectively, according to CA trends over the study period. The road feature changes from manual drawings illustrated a real change of landscape in the Bamako district from 1990 to 2018. Important changes were observed for all types of road feature.

This study revealed the capabilities of landscape indexes to monitor well landscape pattern changes of urban sprawl.

The findings show that urban sprawl is still actually one of the spatial phenomena that highly impacts the urban or natural environment in Bamako district, a deep understanding of these patterns and dynamics is necessary for all future planning or policy actions. Thus, this study suggests further studies on sprawl pattern change and its driving forces. Urban sprawl is a complex and difficult phenomenon to grasp, but this study will make it possible to further understand the process of the phenomenon, and to grasp important details that studies solely on landscape change by indices cannot provide.

**Supplementary Materials:** The following are available online at https://www.mdpi.com/2413-8851/5/1/4/s1, Figure S1: Land use/cover maps for Commune 1: a (1990); b (2000); c (2010); d (2018), Figure S2: Land use/cover maps for Commune 2: a (1990); b (2000); c (2010); d (2018), Figure S3: Land use/cover maps for Commune 3: a (1990); b (2000); c (2010); d (2018), Figure S4: Land use/cover maps for Commune 4: a (1990); b (2000); c (2010); d (2018), Figure S5: Land use/cover maps for Commune 5: a (1990); b (2000); c (2010); d (2018), Figure S6: Land use/cover maps for Commune 6: a (1990); b (2000); c (2010); d (2018), Figure S7: Land use/cover maps for River left side: a (1990); b (2000); c (2010); d (2018), Figure S8: Land use/cover maps for river right side: a (1990); b (2000); c (2010); d (2018).

**Author Contributions:** M.A.K.: Conceptualization, methodology, writing original draft preparation, writing editing; R.R.: Supervision, software, revision, validation; R.A.: Supervision, validation, funding acquisition. All authors have read and agreed to the published version of the manuscript.

**Funding:** Key R & D Program of Jiangsu Province [No. BE2017115].

**Institutional Review Board Statement:** "The study was conducted according to the guidelines of the Declaration of Helsinki, and approved by the Institutional Review Board (or Ethics Committee) of NAME OF INSTITUTE (protocol code XXX and date of approval)."

**Informed Consent Statement:** Informed consent was obtained from all subjects involved in the study.

**Data Availability Statement:** There is no data statement because data used in this article don't need any permission.

**Acknowledgments:** This work was supported by the Key R & D Program of Jiangsu Province [No. BE2017115].

**Conflicts of Interest:** "The funders had no role in the design of the study; in the collection, analyses, or interpretation of data; in the writing of the manuscript, or in the decision to publish the results".

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
