# Peer review of "Spatiotemporal Change of Urban Sprawl Patterns in Bamako District in Mali Based on Time Series Analysis"

_urbansci, doi:10.3390/urbansci5010004_

Round 1

Reviewer 1 Report

The work presents a rather classical work of remote sensing analysis. First of all, I would like to point out a few theoretical points that could be helpful to better position the work. The first is related to the notion of urbanization, pointed out at the beginning without (m)any references, and further ignored declaring that the work will perform a landscape pattern analysis. What is the crucial point in this analysis? Many majors works have been occupied over the last decades (and much beyond), in the deep comprehension of urbanization as a process that deeply transforms territories (Gottmann 1961, the megalopolis; Lefebvre 1970 The urban revolution; Brenner 2014, Implosion Explosion...) in different ways. In my opinion, if one declares the intention of observing ‘Spatio-Temporal Change of Urban Sprawl Patterns’, a minimum positioning within the literature is required. Moreover, in this case, the chosen classes of analysis should try to refer more about the urbanization process and less about landscape patterns, even if disappearing ones, and thus recording a growing urbanisation. It would be interesting indeed, for once, to see a work that distinguish inside this generic ‘build up’ category at least a couple of different appropriation of space. Bamako is a city that has a clear double structure: the planned city, the main equipment along the asphalted roads where the more formal city take place. On the other hand, there is a second layer of informal urbanization, of low-density settlement (this distinction more evident in the south, but rather clear in both side of the river), that have a completely different impact into existing environment: they make use of local resources both for constructing materials and for producing food. I am not asking for a socio-anthropological investigation, but simply pointing out that, the observation could be enriched trying to further categorize (even with different cartographic sources or aerial picture surveys) this built-up space that as a protagonist remains way too generic and thus ignored. The second point of observation is following the same line of thinking and refers to the fact that the work makes indifferent use of land use (LU) and land cover (LC), while these are absolutely not the same. There are – at least- two way of distinguish them: on one side LC could represent the output of an analysis such as the one here proposed, while the LU represent the ‘planned’ activities dedicated to these zones. Even in Bamako is present a zoning plan, which might not cover the same area of analysis, fact that could enrich the discussion, highlighting those developments that are ‘informal’ according to local law. On the other hand, LC could refer to the materiality of the land (grass, forest, built, water etc) and the LU could then refer to the use that the city is making of the same zones. For example, built up could be used for living, or for producing, grass could be a sport field or simply a field… and so on. Other argumentations of this distinction, especially according to the different scales of analysis are possible, but LU and LC might be distinguished, even making use or more detailed scales of analysis, which in my opinion will deliver a much more interesting work.

Moreover, despite the classical methodology and correct use of materials presented the work presents in my opinion some lack of clarity in the -way too few- references proposed. For the whole section of introduction and part of the methodology, the authors refer to study/concepts without quoting the related literature, or missing to refer to sources. Just for making some examples: line 34. how many? what are the positions? which are the references? such a sentence should be supported; line 62: there are also many other type of studies that look at the transformation of landscape patterns with different methods. See, for example: look also at the special issue Vol 5, No 2 (2020) on ‘Urban Planning’ ‘territories in time: mapping palimpsest horizons’; line 77-78; line 108-117, what is the source of this ‘geographical description? What is this description given for?; none of the ‘acquisition dates’ in table 1 are in summer, please revise the sentence or further explain the table). In conclusion I believe that the work could be enriched with much more interesting position and observation, that would require a consistent rework of the presented materials.

Author Response

RESPONSES TO REVIEWER 1 COMMENTS:

Point 1: What is the crucial point in this analysis?

Response 1: The crucial point of analysis is to analyze landscape pattern changes related to urbanization, mainly on urban sprawl patterns changes. We have added other author where We had pointed out urbanization at the beginning that urbanization was explored by many researchers. Please refer to line 30.

We also added a part of literature focused on urban sprawl pattern in so far it is the crucial point of this analysis. Please refer to lines from 80 to 89

About your suggestion to make a work that distinguish inside this generic ‘build up’ category at least a couple of different appropriation of space. We inform you that this work is about the third paper We are writing now in which We done a multi-scale level analysis by analysis sprawl pattern change in each of the six administrative subdivision of Bamako (six municipalities), in each river’ side, and by making comparison between. Thus, that is why your suggestion is not included in the current paper.

Point 2: Point 2: The second point of observation is following the same line of thinking and refers to the fact that the work makes indifferent use of land use (LU) and land cover (LC), while these are absolutely not the same.

Response 2: Thank for your very clear observations. Our analysis is about land cover but not land use. So, We have deleted land use from the paper where it is not the case. Changes are highlighted to lines 14, 97, 173, 207, 235, 236, 246, 251, 297, 439

Point 3: Moreover, despite the classical methodology and correct use of materials presented the work presents in my opinion some lack of clarity in the -way too few- references proposed. For the whole section of introduction and part of the methodology, the authors refer to study/concepts without quoting the related literature, or missing to refer to sources. Just for making some examples: line 34. how many? what are the positions? which are the references? such a sentence should be supported; line 62: there are also many other type of studies that look at the transformation of landscape patterns with different methods. See, for example: look also at the special issue Vol 5, No 2 (2020) on ‘Urban Planning’ ‘territories in time: mapping palimpsest horizons’; line 77-78; line 108-117, what is the source of this ‘geographical description? What is this description given for?; none of the ‘acquisition dates’ in table 1 are in summer, please revise the sentence or further explain the table). In conclusion I believe that the work could be enriched with much more interesting position and observation, that would require a consistent rework of the presented materials.

Response 3: Thank you for your kind comment. We cited some reference to support our citations. Please refer to lines 30, 70, from 80 to 89, 421, 424.

We have made change citing the table 1 which shows the characteristics and details about the used images. Please refer to line 133.

we have made change in introduction part based on your remarks. Sources are cited to supported some citations. Refer to lines from 80 to 89

CLOSING COMMENTS TO REVIEWER 1’s COMMENTS:

Thank you for all your insightful comments. Thank you for taking the time and energy revise the paper. We hope that our revision improved the quality of the paper such that you will deem it worthy of publication.

Reviewer 2 Report

I enjoyed reading this manuscript. Also, I believe that this paper could be a significant contribution to the urban field, especially for the necessity of research about the implications of urban sprawl in developing countries.

I consider that this paper could be published in "Urban Science". However, the authors could improve the paper before their publication:

Figures 5 and 6 need to be improved, according to the format of figures 7 and 8.

One recommendation is that sections 3 and 4 merge into one single section: Results and discussion. I consider that the authors discuss some results in section 3, and section 4 looks poor. Also if authors integrate more references that support their discussions; the paper could increase their scientific quality.

Also, I believe that the authors have very interesting results, so, if they talk about the implications that the fragmentation of landscape has in Bamako (supported by other references); definitely, the paper could have a higher diffusion.

Author Response

RESPONSES TO REVIEWER 2’s COMMENTS:

Point 1: Figures 5 and 6 need to be improved, according to the format of figures 7 and 8.

Response 1: We have improved figure 5 and 6 according to your suggestions. I still keep figure 5 in this format to avoid to get a lot of number of figures in the document. Refer to line 309, 335.

Point 2: One recommendation is that sections 3 and 4 merge into one single section: Results and discussion.

Response 2: Thank you for your recommendation. We have merged results and discussion to one section. Please refer to line 205.

Point 3: Also, if authors integrate more references that support their discussions; the paper could increase their scientific quality.

Response 3: We integrated some references to support our citations. Please refer to lines 30, 70, from 80 to 89, 421, 424.

There is no study about fragmentation of landscape concerning Bamako. This study is experiencing the first one and that is why any reference is cited about Bamako.

CLOSING COMMENTS TO REVIEWER 2’s COMMENTS:

Response: Thank you very much for your kind words about our paper. We appreciate you taking the time to offer us your comments and insights related to the paper. We found your feedback very constructive. We tried to be responsive to your concerns. We hope you find these revisions rise to your expectations.

Reviewer 3 Report

Urbsci990666 Spatio-temporal change of urban sprawl patterns In Bamako District in Mali based on time series analysis

This paper discusses an interesting piece of African urban land, too often neglected from the mainstream research. In the present study, the Authors described four Landsat images from remote sensing. They conclude that urban sprawl highly affects environment in Bamako district. In addition, nevertheless this study, a deep understanding of these patterns and dynamics is necessary for all future planning or policy actions. And that, further study on sprawl patterns change and its driving forces: being urban sprawl being a complex and difficult phenomenon to grasp, this further study will make it possible to understand more the process of the phenomenon and to grasp important details that the only study of landscape change by indices cannot provide.

This manuscript adheres to the journal’s standards. The research meets the applicable standards for the research integrity. The article does not adhere to appropriate reporting guidelines and community standards for data availability: the complete raw database is not yet made completely available in a public repository, such as Zenodo, for instance.

The research output, in terms of novelty, scores extremely poor uniqueness in terms of data, mainly due to the​​ nebulous aim. The level of clarity is below the threshold of acceptability, as well as the state of the art and the comparative discussion. It partially adopts up to date methodologies in respect to the object of research. The paper does not discuss the limitations of the approach and potential biases due to the assumptions made.

The very few papers on this area (e.g., 10.1016/j.landusepol.2018.10.045) must be discussed comparatively, and used as precious point of reference. Metrics, presented here as important novelties, are well known indeed. The Reader, get the impression from this reading that satellite can eliminate the need for field validations. This is not true.

Potentially, its potential impact upon the international scientific community of reference is low. The study presents the results of primary scientific research, while the comparison with other studies described in the literature should be the toughest section, although not well exploited into the Discussion section. Experiments, statistics, and other analyses are performed to a moderately sound technical standard but are described in poor detail. Conclusions presented are not innovative.

The article is presented in a quite intelligible manner. This work has not yet a sufficient impact and does not add yet to the knowledge base.

Title: It must be focused

Keywords: some are a mere repetition of the title. Not effective for retracing the article in a literature search

Abstract: OK. Please, a couple of sentences on implication would be fine.

Introduction: REVISE. Please, delete the whole foggy part on the importance of RS. The aim of this work must contain a new question

Method: REVISE. Please refer to internationally shared methods, reporting differences where necessary

Discussion: REVISE. Strengthen, and re-focus the whole section on the existing recent Literature re-interpretation

Conclusion: REVISE. The conclusions could be stringent, and intriguing

Figures: Please reduce the whole number, more than half

In particular (page.row):

3.103 Please, insert the full geographical coordinates

3.110 Please, refer to IUSS WG WRB (2015)

9.209 This seems to me to be one of the main results

Figure 1. Unnecessary

The manuscript contains several typos in the list of references that does not follow strictly the journal Instructions

Recommendation: REJECT – with option to resubmit

References

IUSS Working Group WRB. 2015. World Reference Base for Soil Resources 2014, update 2015 International soil classification system for naming soils and creating legends for soil maps. World Soil Resources Reports No. 106. FAO, Roma IT EU, 192 p.

Author Response

RESPONSES TO REVIEWER 3’s COMMENTS:

Point 1: 3.103 Please, insert the full geographical coordinates

Response 1: We have added the full geographical coordinates. The change is highlighted to line 115.

Point 2: 3.110 Please, refer to IUSS WG WRB (2015)

Response 2: Please We didn’t understand what you mean here to refers to IUSS WG WRB. We have checked the page 3 line 110 but We didn’t find what you want me to change.

Point 3: Discussion: REVISE. Strengthen, and re-focus the whole section on the existing recent Literature re-interpretation

Response 3: Thank you for your recommandations have revised the discussion and some recent literatures are included. The changes are highlighted to lines 422 to 424. Also we have merged the discussions and conclusion sections to one section according to reviewer 2 suggestion.

Point 4: Abstract: OK. Please, a couple of sentences on implication would be fine.

Response 4: Thank you suggestions. The strict limitations of words number to 200 words in Abstract is the raison that it was not possible for me to maintain in the abstract all its requirements for like implications.

Point 5: Keywords: some are a mere repetition of the title. Not effective for retracing the article in a literature search

Response 5: We have revised the keywords. Please refer to line 26

Point 6: Conclusion: REVISE. The conclusions could be stringent, and intriguing

Response 6: We have also revised the conclusion, refer to line from 451 to 455.

Point 7: The manuscript contains several typos in the list of references that does not follow strictly the journal Instructions

Response 7: Thank you for your comments. We revised the reference.

CLOSING COMMENTS TO REVIEWER 3’s COMMENTS:

Response: Thank you very much for your kind words about our paper. We appreciate you taking the time to offer us your comments and insights related to the paper. We found your feedback very constructive. We tried to be responsive to your concerns. We hope you find these revisions rise to your expectations.

Round 2

Reviewer 1 Report

Dear authors, thank you for taking into consideration the comments of the previous review. Some minor changes have been done to the work, whereas I believe that a much structural review was needed, as argued in the previous review: a broad re positioning of the state of the art; the considering of improving the categories of analysis (although this work is part of a series), and ultimately the questioning of land use and cover which have been solved simply  by substituting the word. 

Author Response

Point 1: Dear authors, thank you for taking into consideration the comments of the previous review. Some minor changes have been done to the work, whereas I believe that a much structural review was needed, as argued in the previous review: a broad re positioning of the state of the art; the considering of improving the categories of analysis (although this work is part of a series), and ultimately the questioning of land use and cover which have been solved simply  by substituting the word.

Response 1: Thank you for your comments. We have change in the paper by adding a part of research abroad as you suggested. Refer to lines from 33 to 43 to see the changes.

Reviewer 3 Report

Previous round of review was totally unattended

Author Response

Point 1: Previous round of review was totally unattended

Response 1: Thank you for your comment dear reviewers. But I am sorry to be not able to satisfy you comment but don’t know exactly what you wanted. Thus, I opt for the rebuttal option about your suggestion.
